



# Synoptic patterns associated with high-frequency sea level extremes in the Adriatic Sea

Krešimir Ruić[1], Jadranka Šepić[1], Marin Vojković[1]

[1]Faculty od science, University of Split, Split, 21000, Croatia

*Correspondence to*: Krešimir Ruić (kruic@pmfst.hr)

**Abstract.** The study focuses on classification of synoptic conditions leading to episodes of extreme high-frequency sea level oscillations in the Adriatic Sea (Mediterranean). Two types of extreme episodes were obtained from sea level time series measured at six tide gauge stations: (i) HF extremes, extracted from high-frequency components (periods shorter than 2

hours) of sea level time series and defined as periods in which high-frequency component was above a threshold value; and (ii) Compound extremes, extracted from residual (de-tided) time series, and defined as periods in which both high-frequency and residual components were above their respective thresholds. Characteristic synoptic situations preceding both types of extremes were determined using the k-medoid clustering method applied on the ERA5 reanalysis data (mean sea level pressure, temperature at 850 hPa, and geopotential height of 500 hPa level). The structural similarity index measure (SSIM)

was used as a distance metric. The data were divided into a training set (from the start of measurements to the beginning of 2018) and a testing set (from the beginning of 2018 to the end of 2020). For each station, k-medoid was used to obtain first 2 and then 3 clusters with characteristic synoptic patterns called medoids. Two distinct patterns, related to HF and Compound extremes were identified at all stations: (i) "summer-type" pattern – characterized by non-gradient mean sea level pressure, warm air advection from the south-southwest at 850 hPa, and a presence of a jet stream at the 500 hPa height, with all three

conditions previously found to favour development of meteorological tsunamis, i.e., the strongest of atmospherically triggered high-frequency sea level oscillations; (ii) "winter-type" pattern characterized by pronounced mean sea level pressure gradients favouring winds which induce storm surges, colder low troposphere, and a presence of a jet stream at the 500 hPa level. Including the third cluster into the analysis led to extraction of either a novel "Bora-type" pattern involving strong northeast winds at stations Bakar and Rovinj, or an additional cluster with a medoid which represents refinement of

summer- or winter-type patterns. The extracted medoids of clusters were used to label all days of the testing period. It was shown that HF or Compound episodes recorded in the testing period mostly appeared during synoptic situations which highly resembled extracted medoids. The potential of using k-medoid method for forecasting high-frequency sea level oscillations is discussed.



## 1 Introduction

Sea level variability manifests itself on time scales from seconds to millennia and on spatial scales from a centimetre to the global scale (Pugh and Woodworth, 2014). In this paper, we focus on high-frequency sea level oscillations (also referred to as a short-period oscillations), which occur on temporal scales of a few minutes to a few hours and on spatial scales of a few to hundreds of kilometres. These oscillations, which include free and forced long ocean waves (Pugh and Woodworth, 2014), edge waves (Ursell, 1952) and seiches (Rabinovich, 2009), can be atmospherically triggered, but can also be associated to tsunamis of seismic, landslide and/or volcanic origin (Pugh and Woodworth, 2014). Herein, we study only atmospherically triggered high-frequency sea level oscillations, the strongest of which are often referred to as meteorological tsunamis (or meteotsunamis) (Monserrat et al., 2006). Meteotsunamis are generated by atmospheric pressure (and wind) disturbances which propagate over the open sea, and which transfer energy to the ocean through a process first described by Joseph Proudman (Proudman, 1929), and later termed Proudman resonance by Mirko Orlić (Orlić, 1980). Such atmospheric disturbances are often related to atmospheric gravity waves (e.g., Monserrat and Thorpe, 1996) but can also be due to convective pressure jumps (Jansà et al., 2007; Belušić et al., 2007), derechos (Šepić and Rabinovich, 2014), and squall lines preceding cold fronts (Pellikka et al., 2022). Intense high-frequency sea level oscillations can also be triggered by wind blowing over a bay or lake or some other limited area (Wilson, 1972).

Many meteotsunamis (i.e., individual events of the most extreme atmospherically triggered high-frequency sea level oscillations) have been analysed in detail – through theoretical studies, atmospheric and ocean data analysis, and numerical modelling (see Rabinovich (2020) for an extensive list of research on the strongest known events). However, statistical analyses of high-frequency sea level oscillations are not so numerous. Using the NOAA dataset, Bechle et al. (2016) and Dusek et al. (2019) performed the first comprehensive statistical analyses of sea level extremes occurring at periods shorter than 6 hours along the Great Lakes and the USA East Coast, respectively, with both studies revealing that these oscillations can pose a significant risk for the coastal area. Šepić et al. (2015a) used the UNESCO Sea level Monitoring Facility database (https://www.ioc-sealevelmonitoring.org/) which contains data measured with 1-15 minutes time step to analyse high-frequency sea level oscillations in the Mediterranean basin and relate them to the prevailing atmospheric conditions. Work of Šepić et al. (2015a) was subsequently expanded by Vilibić and Šepić (2017) and Zemunik et al. (2022a, 2022b) who estimated global distribution of high-frequency sea level oscillations, including estimations of their variances, typical ranges, seasons and other relevant characteristics. Analyses described in listed papers were all performed on high-pass filtered sea level series (cut-off periods of 2 or 6 hours, depending on a study), and thus they assessed only the relevance of high-frequency sea level oscillations "per se". However, recent research has shown that high-frequency sea level oscillations often accompany lower-frequency (periods longer than a few hours) extreme sea level events, for example storm surges, and that they can contribute significantly to total sea level height during such events. Some examples of the events where extreme sea levels and floodings were caused by the joint acting of a storm surge and high-frequency sea level oscillations include: Storm Gudrun in the Baltic Sea (Suursaar et al., 2006), Storm Gloria in the western Mediterranean (Pérez-Gómez et al.,





2021), Typhoons Lionrock and Jebi on the coasts of Japan (Heidarzadeh and Rabinovich, 2021), Typhoon Maysak in Korea

and the Sea of Japan (Medvedev et al., 2022), and Typhoon Songda and related extratropical cyclones in British Columbia

and Washington State (Rabinovich et al., 2023).

Recently, Ruić et al. (2023) made another step forward by analysing the contribution of high-frequency sea level

oscillations (T < 2 h) to total sea level extremes of the Adriatic Sea (Fig. 1). The authors have analysed 1-min sea level

measurements from the eighteen tide gauge stations, with the six longest records having ~17 years of data, and have shown

that high-frequency sea level oscillations can give rise to extreme sea levels of the Adriatic Sea, both independently -

meaning that extreme sea level height is due to high-frequency component only, and jointly with a low-frequency

oscillations (T > 2 h) – meaning that the extreme sea level height is due to combined effect of low-frequency oscillations (T

> 2 h), such as storm surges, and high-frequency oscillations (T < 2 h). In their discussion, the authors note a distinct

seasonal distribution of the strongest high-frequency sea level oscillation, leading them to suggest that these oscillations are

likely linked to specific synoptic weather patterns. Extraction of synoptic patterns related to the extreme high-frequency

oscillations in the Adriatic Sea is the topic of our work.



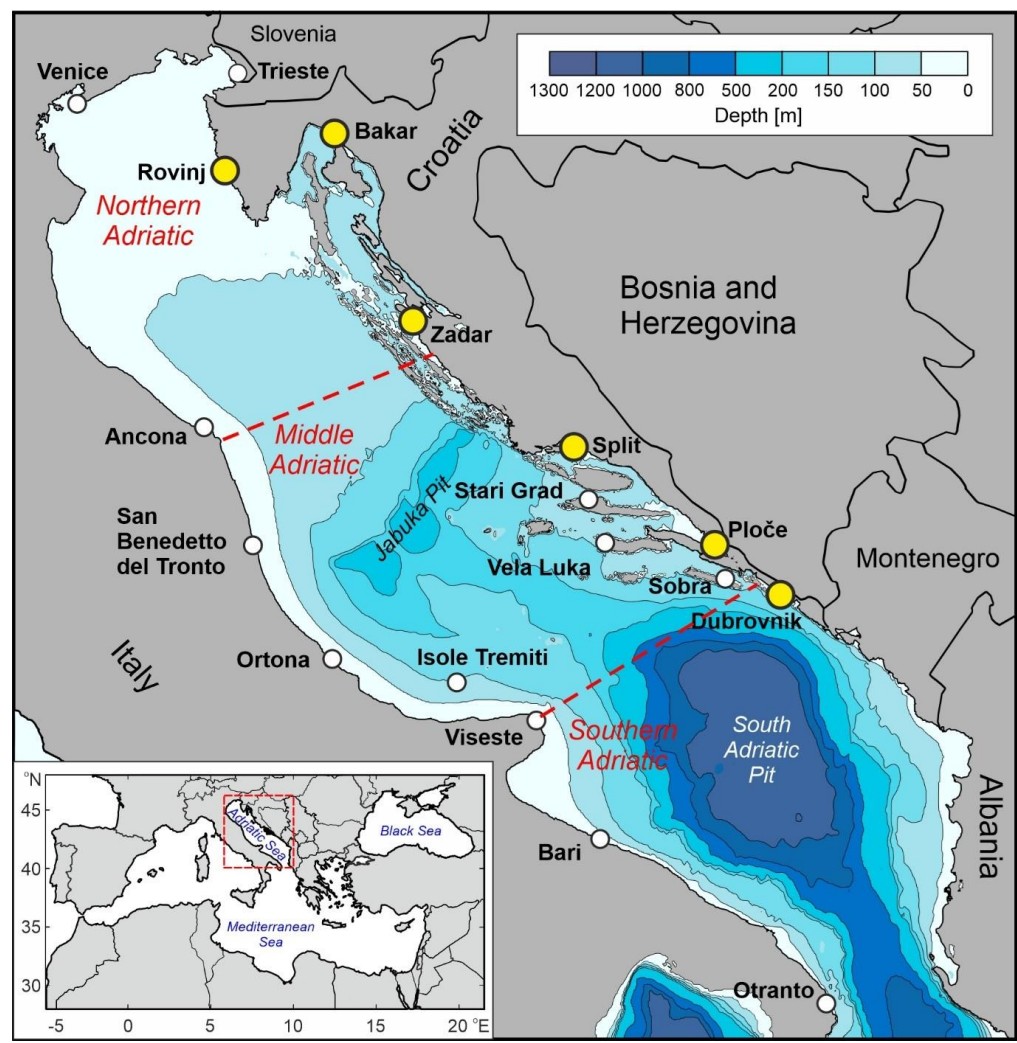

**Figure 1. Bathymetry, locations, and names of tide gauge stations (circles) used in Ruić et al. (2023). Coloured yellow circles mark tide gauges analysed in this paper.**

Numerous studies, particularly for the Mediterranean, have shown that the strongest atmospherically induced high-frequency sea level oscillations generally occur when distinct synoptic conditions prevail over the area. The pioneering studies go back to Ramis and Jansà (1983) for the Balearic Islands and to Hodžić (1988) for the Adriatic Sea. The characteristic conditions over the Mediterranean include a three-layer troposphere with a well-mixed, warm and moist shallow surface layer extending to an altitude of ~900 hPa. This layer is overlain by a temperature inversion, followed by a

deeper layer, which is warm and dry in its lowest part, but whose temperature decreases with a high rate, and whose humidity and wind speed increase with altitude, possibly leading to conditionally or dynamically unstable mid and upper troposphere layers. On the horizontal scale, a low-pressure trough is often found to the west of the area where high-





frequency sea level oscillations occur, warm air is advected from the southwest at altitudes higher than 900 hPa, and a front size of a deep upper-level trough, associated to strong mid and upper troposphere south-westerly winds, is located right over

the area. In the following decades, other authors documented similar favourable synoptic conditions for the Balearic Islands, Adriatic Sea and other Mediterranean locations (Jansà et al., 2007; Vilibić and Šepić, 2009; Šepić et al., 2009; Šepić et al., 2015a; Šepić et al., 2015b). However, more recent research has shown that Mediterranean meteotsunamis also appear under different synoptic conditions, i.e., in situations dominated by deep extratropical cyclones which can, through joint action of pressure drop and onshore winds, generate storm surges (Ferrarin et al., 2021; Šepić and Orlić, 2024). The extraction of

synoptic patterns leading to strong high-frequency sea level oscillations has also been carried out for a couple of other worldwide locations using mostly subjective (visual) techniques (e.g., the Great Lakes coast, Bechle et al., 2016; the Baltic Sea, Pellikka et al., 2022). At these locations several characteristic synoptic patterns have been recognised.

Herein, we aspire to classify and separate synoptic situations which lead to episodes of extreme high-frequency sea level oscillations from those leading to episodes of simultaneous extreme high- and low-frequency sea level oscillations, all

for the Adriatic Sea. As already noted, our work builds on paper by Ruić et al. (2023), i.e., on analysis on synoptic patterns which prevailed in the atmosphere during ~300 episodes of high-frequency sea level extremes extracted by Ruić et al. (2023) from ~17 years of 1-minute sea level measurements at six Adriatic Sea tide gauges. We perform the classification by applying the k-medoid clustering method (Kaufmann and Rousseeuw, 1987) to the ERA5 reanalysis fields (Hersbach et al., 2020). The k-medoid algorithm, which groups objects to a predetermined number of clusters using a selected distance metric

(in our case SSIM) is explained in more detail in Materials and methods. The paper is structured as follows: Section 2 brings materials and methods; in Section 3 results are presented and in Section 4 the discussion and conclusions are given.

## 2 Materials and methods

Ruić et al. (2023) analysed sea level series measured with a 1-minute time step at 18 tide gauge stations located along the eastern and western coasts of the Adriatic Sea (Fig. 1). Length of the time series was 3-17.5 years. Prior to the

analyses, the authors performed rigorous quality control of all sea level series, removing of all non-physical spikes and outliers. Following this, the authors de-meaned, and de-tided the series. The series were de-tided using the MATLAB T-TIDE package (Pawlowicz et al., 2002), with tidal signal estimated for the seven constituents which are traditionally used for the Adriatic Sea tidal analyses, namely for K1, O1, P1, K2, S2, M2 and N2 (Kesslitz, 1919). Afterwards, the residual sea level series (series without the tidal components and with subtracted mean values) were filtered with a 2-hour Kaiser-Bessel

window (Thomson and Emery, 2014) resulting in the formation of high-frequency (HF, T < 2 h) and low-frequency (LF, T > 2 h) series. The authors defined extremes on HF series using the peak-over-threshold method, defining all points in the HF time series that are above the 99.993 percentile threshold as extremes. These extremes were called "High-frequency" extremes ("HF extremes", from this point on). Additionally, authors have noticed that there are situations when HF extremes occur within +-24 hours of extremes of residual series (termed as Residual extremes in Ruić et al., 2023) with the latter



defined as periods when residual sea level surpasses its 99.85 percentile. Herein, we term these "joint" events "Compound extremes". For both types of extremes (HF and Compound), to ensure the independence of events, a condition was set, that all points which surpass the percentile thresholds, and which appear within a 3-day window represent the same extreme episode. As an example, in Fig. 2 we show one HF extreme and one Compound extreme measured at the tide gauge Bakar (Fig. 1).

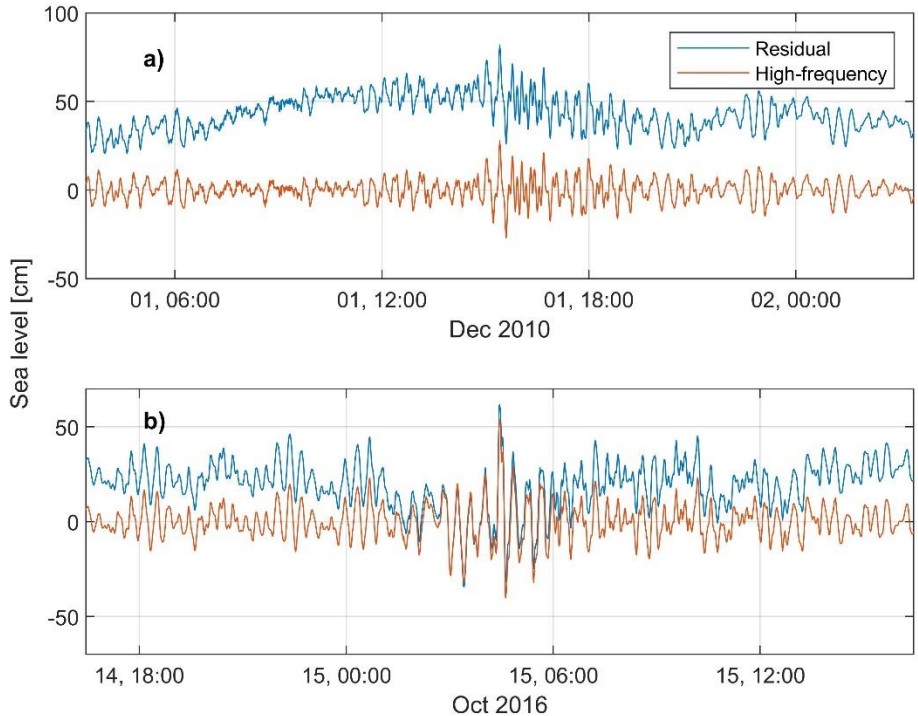


**Figure 2: a) Compound extreme of 1 December 2010; b) HF extreme of 15 October 2016.**

In this paper we classify synoptic situations which govern appearance of HF and Compound extremes at 6 tide gauge stations (out of 18 stations used by Ruić et al., 2023). These six stations (Rovinj, Bakar, Zadar, Split, Ploče,
Dubrovnik; Fig. 1) have the longest time series (16.4-17.9 years) and are evenly spread along the eastern coast of the Adriatic Sea. Unfortunately, stations located along the western Adriatic Sea coast possess time series of too short total duration (from 6 to 10 years) to allow for extraction of a sufficient number of extreme episodes needed for applying clustering methods. For the selected six stations, information on the HF and Compound extremes (date, heights of residual, HF and LF component, for each event) were extracted from the paper or from data used by Ruić et al. (2023). Basic
properties of HF and Compound extremes at each station are listed in Table 1.



Data availability period was divided into two parts: the training and the testing period. The training period spans from the start of the measurement (slightly different for each tide gauge, at the earliest: 1 January 2003; at the latest: 19 June 2003), until 31 December 2017 and the testing period from the 1 January 2018 until 31 December 2020.



**Table 1. Locations of tide gauges, length of time series (in years), as well as percentile thresholds (in cm) for defining**
**HF and Compound extremes and the number of both types of episodes (HF and Compound) in the testing and training periods.**

| Tide gauge | Location | Length of series [years] | Percentile threshold for HF; Compound extremes [cm] | No. HF/ Compound in training period (2003-2017) | No. of HF/ Compound in testing period (2018-2020) |
|---|---|---|---|---|---|
| Dubrovnik | 42.65 N, 18.09 E | 17.1 | 5/43.2 | 28/6 | 11/3 |
| Ploče | 43.05 N, 17.43 E | 17.9 | 18.4/46.7 | 33/6 | 5/3 |
| Split | 43.51 N, 16.44 E | 17.4 | 10/47.6 | 40/5 | 9/2 |
| Zadar | 44.12 N, 15.24 E | 17.5 | 10.9/51.6 | 37/7 | 7/3 |
| Bakar | 45.37 N, 14.62 E | 16.4 | 21.4/62.4 | 50/6 | 5/3 |
| Rovinj | 45.08 N, 13.63 E | 16.8 | 11.1/62 | 27/3 | 15/4 |

Classification of synoptic conditions was performed by applying the k-medoids clustering method to the ERA5
reanalysis data (Hersbach et al., 2020; Copernicus Climate Change Service (C3S), 2017). After initial tests with a wider range of variables, we focused on the following ones: (i) mean sea level pressure (MSLP); (ii) temperature at 850 hPa; and (iii) geopotential height at 500 hPa. Herein, we note that favourable conditions for generation of strong high-frequency sea level oscillations can usually be detected in spatial fields of these synoptic variables (e.g., Jansà et al., 2007; Vilibić and Šepić, 2009; Šepić et al., 2015b). The variables were downloaded for the area of the Adriatic Sea (approximate area shown
in Fig. 1) for 12:00 UTC of those days of the training period on which HF or Compound extremes occurred and for 12:00 UTC of each day of the testing period.

For each ERA5 variable, means and standard deviations of each month were calculated using the ERA5 data for the whole period (from 2003 until 2021). Synoptic data corresponding to each HF and Compound extreme of the training set and to each day of the testing set were then normalized by subtracting the monthly mean and dividing the resulting series by the
monthly standard deviation. Normalization was done for each variable separately. All normalized variables (MSLP, temperature at 850 hPa and geopotential height at 500 hPa) were concatenated to form a single vector (called data point) for each day of the training period in which a HF or a Compound extreme occurred, and for each day of the testing period. Characteristic synoptic clusters were then obtained by applying the k-medoids algorithm on all vectors from the training period. Clustering is normally done by first selecting a number of clusters ($n$) and then iteratively evaluating similarity of
input vectors to each other. The process of evaluating is repeated until $n$ input vectors (i.e., $n$ vectors of synoptic variables related to one of the HF or Compound extremes) most alike to all other input vectors within a cluster are found. The vectors



which are most alike to all other inside a cluster (one per each cluster) are called *medoids*, and basically represent normalized synoptic situations (MSPL; temperature at 850 hPa, and geopotential height at 500 hPa) on one of the training set days.

165   To determine mediods, and group all input vectors into clusters, we have used the structural symmetry index measure (SSIM) as a distance metric. SSIM was calculated as in Wang et al. (2004). If we take for example, temperature, and look at two different days in which a HF extreme occurred, setting $x$ to be the vector of temperature for the first date and $y$ the vector of temperature for the second date, the SSIM value (for temperature) is computed as:

$$SSIM(x,y) = \frac{(2\mu_x\mu_y + c_1)(2\sigma_{xy} + c_2)}{(\mu_x^2 + \mu_y^2 + c_1)(\sigma_x^2 + \sigma_y^2 + c_2)}, \tag{1}$$

where $\mu_x$ and $\mu_y$ are mean values of $x$ and $y$ respectively (in our case both equal 0, as series were normalised) and $\sigma_x$ and $\sigma_y$

170   their variances while the $\sigma_{xy}$ is the covariance. Herein, $c_1$ and $c_2$ are the stabilization coefficients that are necessary in situations in which the denominator (without $c_1$ and $c_2$) in (1) is close to zero. The stabilization coefficients are computed as $c_{1,2} = (K_{1,2}L)^2$, where $L$ is the dynamic range of pixel values and $K_{1,2}$ are small ($<< 1$) constants (see Wang et al., (2004) for further details on computation of $C_{1,2}$, dynamical range, and $K_{1,2}$). The SSIM between two data points (vectors of synoptic variables) is calculated as the average of SSIM for temperature, mean sea level pressure, and geopotential. This average

175   SSIM is referred to as the SSIM value between two data points. These SSIM values, between all pairs of dates in the training set, are then passed as the distance matrix to the k-medoids algorithm. From that distance matrix, the method choses the $n$ medoids as the days with the maximal similarities to all other days.

  Silhouette and Elbow methods were used to determine optimal number of clusters for each station. Silhouette method is used to assess the quality of the preformed clustering in dependence on the number of clusters (Rousseeuw, 1987).

180   For each data point in each cluster, the average SSIM value is calculated between that point and all other data points in that cluster (denoted as $a$). Then, for the same data point the average SSIM value between that point and all other data points in the next nearest cluster is calculated (denoted as $b$). The value of the Silhouette score is then given by: $(b - a)/\max(a,b)$. Higer values indicate better matching of data point to its assigned cluster and worse matching to the neighbouring cluster. A total silhouette score is obtained by estimating the mean value of all Silhouette scores.

185   Another way of assessing the optimal number of clusters is the Elbow method (Kodinariya and Makwana, 2013). The sum of square SSIM values (called inertia here) is computed between each data point and the assigned medoid, within each cluster. Adding more clusters results with decrease of inertia. If there are, e.g., $n$ characteristic situations, the inertia will become drastically smaller after these $n$ situations are extracted. Once these situations are found, the increase in cluster number doesn't contribute significantly to further reduction of inertia value.

190   Figure 3 shows the results of applying Silhouette and Elbow methods to our dataset. Silhouette method (Fig. 3a), shows that, for most stations, highest scores are achieved when 2 to 4 clusters are used, with the optimal cluster number being the one with the highest Silhouette score. For Rovinj, Zadar, Ploče and Dubrovnik the highest score is obtained when two clusters are used. For Bakar, three clusters are the best choice, and for Split, the choice of three or four clusters gives





approximately the same score. As for the Elbow method, we are looking for a point at which inertia value is low, and for
195  which addition of new clusters does not decrease inertia value significantly. For all our stations, the elbow point is located

somewhere between 2 and 5 clusters (Fig. 3b). Given the results of the Silhouette and Elbow methods, we have chosen to

apply k-medoids method setting the number of clusters first to 2, then to 3, and then to compare results obtained with these

two choices.

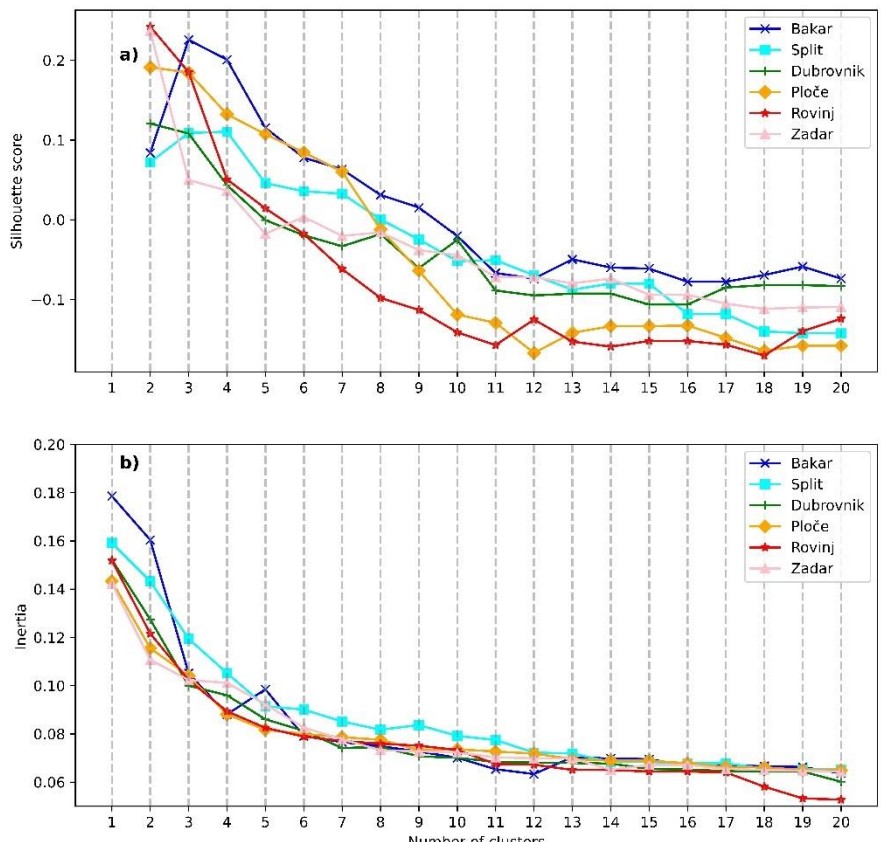

200  **Figure 3: a) Silhouette and b) Elbow methods for determining the number of clusters. Color-coded are the different tide gauge
stations.**



## 3 Results

### 3.1 HF and Compound extremes

As previously stated, we differentiate two types of extremes episodes (previously extracted by Ruić et al., 2023): (1) HF extremes - episodes in which high-frequency sea level oscillations reached extreme values, and (2) Compound extremes - episodes in which both high-frequency and residual series reached extreme values. Yearly distribution and heights of both types of extremes are shown in Fig. 4. At some tide gauges, e.g., Bakar and Zadar events are evenly distributed through the years. At other stations, e.g., Rovinj, events are more abundant in one period (2016-2021) and are rare in other periods (2004-2014). It can also be noticed that some events were captured only at one tide gauge station, while others were captured at multiple tide gauges, for instance the Compound extreme of 29-30 October 2018 was recorded on stations Rovinj, Bakar, Split, Ploče and Dubrovnik.

The maximum sea level of HF and Compound extremes is strongly station dependant, as is the contribution of HF component to the total level. At Dubrovnik (a station with the weakest HF signal), maximum residual sea level during HF extremes ranges from 5 to 52 cm, and at Bakar (a station with the strongest residual signal) from 4 to 134 cm (Fig. 4), whereas HF component ranges from 5 to 14 cm at Dubrovnik, and from 21.4 to 54 cm at Bakar. The residual sea level height during Compound extremes ranges from 34 to 52 cm at Dubrovnik, and from 58 to 134 cm at Bakar, with the contribution of HF component from 6 to 12 cm at Dubrovnik, and from 24 to 51 cm at Bakar. In Rovinj, Bakar, Zadar and Ploče, the HF component occasionally contributed with more than 50% to Compound extremes. On the other hand, in Split and Dubrovnik, this contribution was lower than 30% for all joint episodes.





**Figure 4: Temporal distribution of high-frequency sea level extremes at six tide gauge stations. Blue bars denote HF sea level during HF extremes, red bars denote HF sea level during Compound extremes, and grey bars denote residual sea levels for both types of extremes. The grey shaded area (2018 to 2021) marks the testing period, while the white area marks the training period (2003 to 2018). Red dashed line marks the threshold of the residual sea level height for the definition of a Compound extreme, while the blue dashed line marks the threshold of the HF sea level height for the definition of HF extremes.**





Monthly distribution of two types of extreme episodes is shown in Fig. 5. At Rovinj and Zadar there is a clear seasonal signal with most of HF extremes recorded from June to October. Similar distribution is present for Ploče and Bakar, but with comparable number of events in May as well. In Split, HF extremes peak from April to June, and then in August

and November. At Dubrovnik, the number of HF extremes is even throughout the year – with an exceptionally low number of HF extremes registered in August – this contrasts with most other stations at which highest number of events is registered in August. Regarding Compound extremes, at stations Bakar, Split and Dubrovnik, they are clearly more abundant in colder part of the year (October-December; and less January-April), in line with the known distribution of storm surges (Lionello et al., 2012; Ruić et al., 2023). At Rovinj, Zadar and Ploče, seasonal distribution of Compound extremes is not as pronounced.

However, it should be noted here that at these three stations Compound extremes are occasionally dominantly due to HF component, i.e., during certain episodes HF component is almost high enough to represent a Compound extreme per se (even without exceptionally high LF component) (Fig. 4). If we were to omit such episodes from Compound extremes, the latter would be most abundant in colder part of the year at Rovinj, Zadar and Ploče, as well (not shown).





**Figure 5: Monthly distribution of the HF (orange bars) and Compound (blue bars) extremes at six tide gauge stations, estimated for the entire period of measurements (2003 to 2021).**

Different seasonal distribution of HF and Compound extremes hints that there are at least two different synoptic situations that can produce strong HF oscillations, one associated with the summer-time conditions (presumably similar to






"summer-type" or "good-weather" meteotsunami favourable synoptic situations, as described in Rabinovich, 2020; Pellikka
et al., 2022; Lewis et al., 2023) and the other associated with the fall/winter conditions (presumably similar to "winter-type"
or "bad-weather" meteotsunami synoptic situations, as also described in Rabinovich, 2020; Pellikka et al., 2022; Lewis et al.,
2023).

## 3.2 Characteristic synoptic patterns

Denormalized medoids, i.e., representative situations for each cluster are presented in Figs. 6-8 (in panes "a)" for a
choice of two, and in panes "b)" for a choice of three clusters.  Medoids are shown for Bakar, Split and Dubrovnik which are
selected as representative stations for the north, middle and south Adriatic, respectively. Characteristic medoids for the other
three stations (Rovinj, Zadar and Ploče) are given in Supplementary material (Fig. S1-S3). In all plots, MSLP is shown in the
first row, temperature at 850 hPa in the second row, geopotential height at 500 hPa in the third row and in the final, forth
row, boxplot of HF and LF sea level heights for episodes assigned to each cluster.

Figure 6 shows the results of the k-medoid clustering for Bakar. In accordance with the results shown in Fig. 3, the
optimal number of clusters for classification of synoptic situations related to Bakar HF and Compound extremes is three. The
comparison with the medoids obtained for a choice of two clusters justifies this assessment. A choice of two clusters results
with two similar medoids, while a choice of three clusters produces three different synoptic situations.  Focusing firstly on a
choice of two clusters: the medoid of the first cluster (Cluster A1) had a uniform (non-gradient) MSLP filed over the
Adriatic, while the Cluster A2 medoid was characterized by weak MSLP gradients over the Adriatic Sea favouring weak
southeasterly Sirocco wind. Gradients were due to a closed low located over the Bay of Genoa. The two medoids differed
even less at the higher levels. The temperature fields, for both dates, revealed an inflow of warm air from the southwest in
the low troposphere (850 hPa), while the parallel isohypses indicated that strong south-westerly mid tropospheric winds (jet
stream) were blowing at the height of 500 hPa. Both the jet stream and the inflow of warm air are known precursors to the
generation of intense HF sea level oscillations in the Mediterranean (e.g., Šepić et al., 2015a). The similarity of both medoids
is again confirmed from box plots shown in the fourth row of Fig. 6a where distributions of HF and LF sea level heights
during extreme events associated with a specific cluster are given. These distributions are alike for both clusters, and a
similar number of HF extremes, and a same number of Compound extremes are grouped into each of the two clusters.

For a choice of three clusters, the first medoid (Cluster B1) is alike to the two medoids obtained for a choice of two
clusters. This refers both to synoptic situation and distribution of HF and LF sea level heights during extremes. The Cluster
B2 medoid, on the other hand, represents a new synoptic situation. The surface situation, with an extratropical cyclone
centred over the Tyrrhenian Sea, favoured north-easterly to northerly winds over the northern Adriatic (i.e., Bora, Grisogono
and Belušić, 2009), and Bora, which is characteristic for the Adriatic Sea has, to the best of our knowledge, not previously
been associated to intense high-frequency sea level oscillations. At 850 hPa west-to-east temperature gradient was evident,
and at 500 hPa a closed low was located over the Tyrrhenian Sea, resulting in easterly winds of ~10 m/s over the northern



Adriatic. The third medoid (Cluster B3) shows a situation in which the MSLP distribution favoured southeasterly Sirocco wind and likely a development of a storm surge in the northern Adriatic. There was an inflow of warmer air from the southwest at 850 hPa, and at 500 hPa ispohypses were densified and oriented in a way which implies blowing of strong mid

tropospheric south-westerly winds. Choosing three clusters (instead of two) resulted with clearer separation of episodes according to the maximum sea level height of LF component: median sea level heights and number of Compound extremes are largest for Cluster B3 indicating that this cluster represents situations favourable for development of Compound extremes (with high contributions of both HF and LF components).

In summary, for station Bakar three distinct types of synoptic condition that favour intense HF oscillations (the

strongest being meteotsunamis) can be distinguished: (i) the classic "summer-type" medoid (Clusters A1 and B1) similar to the Mediterranean meteotsunami favourable patterns (Jansà et al., 2007; Šepić et al., 2015a), (ii) (ii) the "Bora-type" medoid (Cluster B2), and (iii) the storm-surge medoid, or "winter-type" pattern (Cluster B3).

**Figure 6: Medoids for a choice of a) two and b) three clusters for Bakar. The first three rows show: MSLP, temperature at 850**

**hPa (T850), and geopotential height at 500 hPa (Z500). Forth row shows box-plots of HF (blue box) and LF (orange box) sea level heights during HF and Compound extremes. The dates of medoids are given at the top of each column and the location of Bakar**





**tide gauge is marked with a circle. Additionally, at the bottom of each column the numbers of HF and Compound extremes grouped in each cluster are given.**

Switching our attention to the middle Adriatic, Fig. 7 shows medoids obtained for station Split for a choice of two

and three clusters. As shown in Fig. 3, the optimal number of clusters for Split is three, however already with two clusters we obtain separation of HF and Compound extremes. Clusters A1 and A2 show two synoptic situations which differed at all studied levels. At the surface layer, both MSLP fields suggested presence of a cyclonic pattern over the Mediterranean. However, isobares of the Cluster A1 median were oriented in a way which led to easterly to northeasterly winds over the Adriatic Sea, whereas isobares of the Cluster A2 median were oriented in a way which led to south-easterly Sirocco winds –

with the latter winds favourable for increase of the LF component. At 850 hPa, a northeast-to-southwest temperature gradient was evident over the Adriatic for Cluster A1, and for Cluster A2, a northwest-to-southeast gradient. The isohypses for both clusters revealed that strong mid-tropospheric winds were blowing in both situations. However, the isohypses (and related winds) were differently orientated, with west-north-westerly winds in Cluster A1, and south-westerly in Cluster A2. Box-plot distributions of HF and LF sea level heights (forth row in Fig. 7a) show that the median of LF sea level heights was

larger in Cluster A2, and that all Compound extremes were grouped in Cluster A2.

Looking at the three-cluster choice (Fig. 7b), the first two medoids (Clusters B1 and B2) are very similar to already described "summer-type" medoid extracted for Bakar (Fig. 6b; Cluster B1) while the medoid of Cluster B3 resembles "winter-type" medoid extracted for Bakar (Fig. 6b; Cluster B3). Thus, unsurprisingly, out of all the HF episodes, the ones with the highest LF values were grouped in Cluster B3, as were all Compound extremes.

In summary, two characteristic and one transitional synoptic situations for generation of strong HF oscillations in Split were extracted, one which is related to "summer-type" (meteotsunami) pattern (Clusters A1, B1 and B2) one which is related to "winter-type" (storm-surge) pattern (Cluster B3), and one "transitional" which has characteristic of both "summer-type" (temperature field at 850 hPa) and "winter-type" (MSLP distribution) pattern (Cluster A2).



**Figure 7: Medoids for the choice of a) two and b) three clusters for Split. The first three rows are: MSLP, temperature at 850 hPa (T850), and geopotential height at 500 hPa (Z500). Forth row are the box-plots of HF (blue box) and LF (orange box) heights during HF extremes assigned to each medoid. The dates of medoids are given at the top of each column and the location of Split tide gauge is marked with a circle. Additionally, at the bottom of each column the numbers of HF and Compound extremes labelled in each cluster are given.**

For the southern Adriatic, medoids for a choice of two and three cluster are shown for station Dubrovnik (Fig. 8). Focusing on pane a) we can see that the medoids for Clusters A1 and A2 both have uniform (non-gradient) MSLP fields over the Adriatic Sea. At 850 hPa, medoid of Cluster A1 shows the advection of warm air from the southwest, whereas a north-to-south temperature gradient was present over the area for the medoid of Cluster A2. Isohypses at 500 hPa had different orientations for Clusters A1 and A2 (third row in Fig. 8a), with south-westerly flow favoured in Cluster A1, and westerly-to-west-north-westerly in Cluster A2. The two medoids are clearly separated by contribution of LF component to extremes. LF component was significantly higher for events belonging to Cluster A2. Additionally, five out of six Compound extremes were grouped in Cluster A2.

Looking at the Fig. 8b., in which the medoids for a choice of three clusters are shown, we can see that the first two medoids (Clusters B1 and B2) are the same as the medoids for a choice of two clusters. This indicates that these two



medoids represent the training set of events so well that even the increase in cluster number does not lead to their change. The third medoid (Cluster B3) had a low-pressure centre located over the middle Adriatic resulting in alongshore and onshore winds at Dubrovnik coast. At 850 hPa there was again advection of warm air from the southwest, and at 500hPa isohypses were oriented in southwest-northeast direction, with south-westerly winds with speeds of ~30 m/s blowing over the southern Adriatic (not shown). Box-plot distribution of HF and LF sea level heights during the extremes reveals that

Cluster B3 contains episodes with the highest LF signal. This cluster has the least HF extremes (4/28) attributed to it but the most Compound extremes (3/6).

In summary, there are three characteristic situations for occurrence of strong HF oscillations in Dubrovnik: (i) one which is the classic "summer-type" (Clusters A1 and B1); (ii) the "transitional-type" one (Clusters A2 and B2); and (iii) the "winter-type" one (Clusters A2 and B3).

Characteristic medoids for the other three stations, Rovinj, Zadar and Ploče, are given in the Supplementary material. At these three station as well, the obtained medoid patterns can be classified as: (i) "summer-type" (uniform MSLP, advection of warm air from the south-west in the lower troposphere and strong south-westerly winds in the  mid troposphere; lower LF component); (ii)  "winter-type" (MSLP gradients favouring sirocco winds which can induce storm surges; colder lower troposphere, and strong south-westerly winds in the mid troposphere; higher LF component, more Compound

extremes); (iii) "transitional-type" medoid (at Zadar and Ploče), and (iv) "Bora-type" (in Rovinj).





**Figure 8: Medoids for the choice of a) two and b) three clusters for Dubrovnik. The first three rows are: MSLP, temperature at 850 hPa (T850), and geopotential height at 500 hPa (Z500). Forth row are the box-plots of HF (blue box) and LF (orange box) heights during HF extremes assigned to each medoid. The dates of medoids are given at the top of each column and the location of Dubrovnik tide gauge is marked with a circle. Additionally, at the bottom of each column the numbers of HF and Compound extremes labelled in each cluster are given.**

Figure 9 shows, separately for a choice of two and three clusters, monthly distributions of the number of HF and Compound extremes within different clusters. Starting with a choice of two clusters, at stations Rovinj, Zadar, Ploče and Dubrovnik, the medoids associated with cluster A1 are more common in summer months (June to August) while the medoids associated with cluster A2 are more common in the colder part of the year. This agrees with the "summer-type" vs. "winter-type" classification. At Bakar and Split Clusters A1 and A2 appear almost evenly throughout the year – confirming that a choice of two clusters did not result with an optimal extraction of synoptic situations at these two stations.

Changing the cluster number from two to three (Fig. 9b), we now notice that "summer-type" (Clusters B1) and "winter-type" (Clusters B3) clusters are separated at all stations, The second cluster (Cluster B2), on the other hand, either: (i) represents refinement of the "summer-type" cluster (in Zadar); or (ii) "winter-type" cluster (Ploče and Durovnik); (iii)




introduces a new (also winterish; peaking in January and February) Bora synoptic situation (Rovinj and Bakar; Fig. 6b; Fig. S1); or (iv) is now associated with spring and autumn events (Split).

Observed seasonal distributions are in line on what is known with climatology of weather patterns over Croatia: calm weather conditions are predominant in summer, extratropical cyclones leading to storm surges peak in autumn but also happen in winter (December to February) (Lionello et al., 2012), and Bora conditions peak in winter, i.e., December to February, but also can happen in autumn (Poje, 1992).

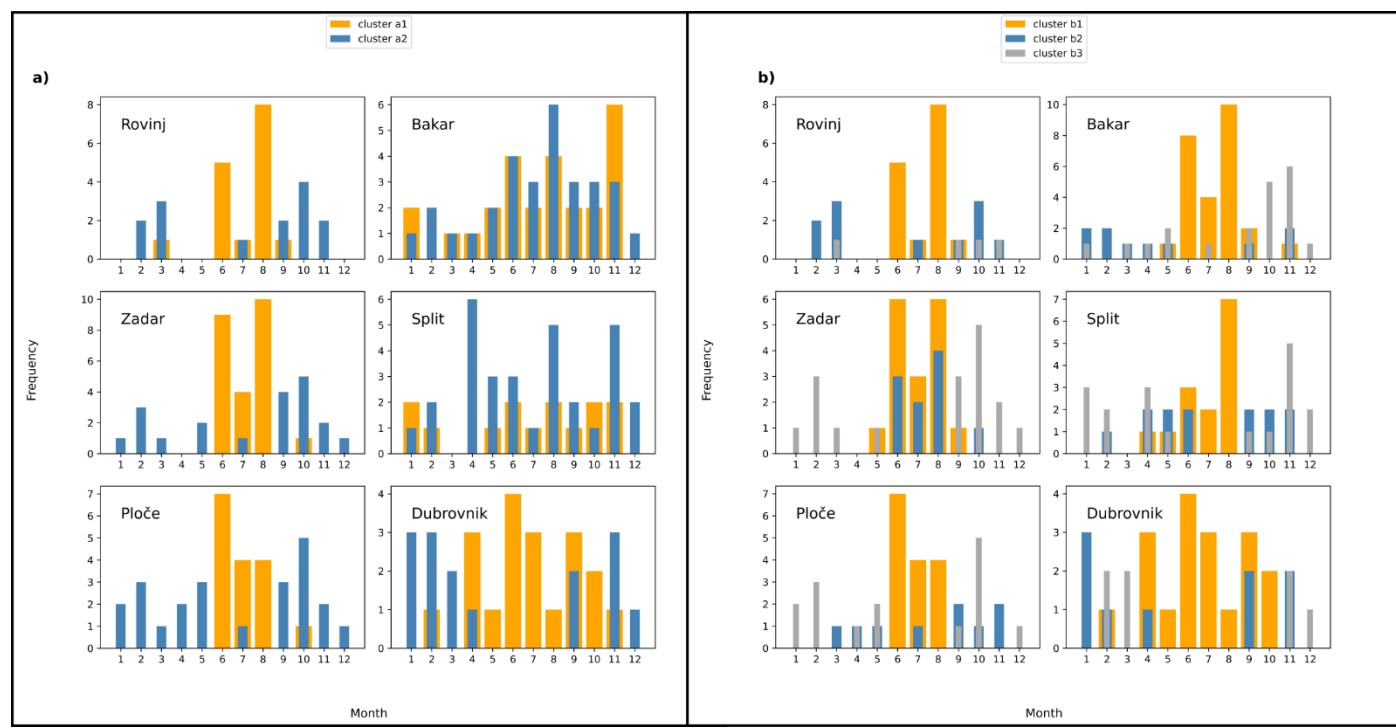

**Figure 9: Monthly distribution of cluster labels for each event in the training part of the data set.**

## 3.3 Testing period

After classifying the HF and Compound extremes from the training dataset into their representative clusters, we tested the classification by feeding the k-medoid algorithm with input vectors (data points) corresponding to selected synoptic variables of 1096 days of the testing period (days from the 1 January 2018 until 31 December 2020). For each day of the testing period, and for each station, and first for a choice of two and then of three clusters, daily synoptic situations (concatenated to vectors consisting of normalised fields of: MSLP, temperature at 850 hPa, and geopotential height at 500 hPa for 12:00 UTC of each day) were compared to characteristic medoids (as extracted from training period, Figs. 6-8 and 1S-3S). Each day of the testing period was assigned to one of the clusters, depending on the SSIM value, i.e., on difference of synoptic situation of that day and synoptic situation of representative medoids, with a primary aim of checking whether



SSIM values were higher on days in which HF or Compound extremes were registered (higher SSIM values suggest higher similarity between fields) than on random days.

Box-plots of the resulting SSIM distributions for each day, cluster and station of the testing period are shown in Fig.
10. In all plots, black dots represent SSIMs for days of the testing period in which an extreme HF or Compound event occurred (based on data used by Ruić et al., 2023). As can be seen in Fig. 10, most SSIM values were above the 75$^{th}$ percentile value (upper edge of boxes) during HF and Compound extremes of the testing period, and all (but 4) above the median value of their respective medoid. At some stations, a choice of two clusters (vs. three clusters) resulted with a better classification of events in the testing period. This holds for Zadar and Rovinj, where both the SSIM median of HF and
Compound extremes, and the percentage of events with SSIM above the 75th percentile decreased when we increased number of clusters from 2 to 3 (Tabe 2). At other stations (Dubrovnik, Ploče and Bakar) the SSIM median increased (implying better classification), but the percentage of events with SSIM above the respective 75$^{th}$ percentile decreased (implying worse classification) when we increased number of clusters. Finally, increasing number of clusters is clearly beneficial for Split station, where both the SSIM median and the percentage of events with SSIM above the corresponding
75$^{th}$ percentile increased.

At Zadar there is one episode that could be considered an "outlier" (Fig. 10). Regardless of the number of clusters, this episode "scores" very low. The related synoptic situation was characterised by exceptionally strong Sirocco winds blowing over Zadar, a uniform temperature field at 850 hPa and a closed low over the Ionian Sea detectable at 500 hPa, differing significantly from extracted Zadar medoids (Fig. 2S).
It is important to note here that the number of events in the testing periods ranges from 5 to 15 for HF and from 2 to 4 for Compound extremes (Table 1.) and that longer period measurements would be beneficial for obtaining more robust statistics.





**Table 2. Values of the SSIM median for all days of the testing period (regardless of the cluster to which these days were assigned), the SSIM median of all episodes of HF and Compound extremes, and the percentage of all HF and Compound extremes above the corresponding 75th percentile, for a choice of two and three clusters.**

| Tide gauge | Two cluster choice | | | Three cluster choice | | |
|---|---|---|---|---|---|---|
| | SSIM median | SSIM median for HF and Compound extremes | Percent of HF and Compound extremes above the 75th percentile | SSIM median | SSIM median for HF and Compound extremes | Percent of HF and Compound extremes above the 75th percentile |
| **Dubrovnik** | 0.574 | 0.673 | 64 | 0.603 | 0.682 | 57 |
| **Ploče** | 0.543 | 0.729 | 75 | 0.548 | 0.746 | 64 |
| **Split** | 0.521 | 0.717 | 64 | 0.528 | 0.743 | 82 |
| **Zadar** | 0.561 | 0.76 | 80 | 0.522 | 0.741 | 60 |
| **Bakar** | 0.485 | 0.722 | 87 | 0.579 | 0.748 | 75 |
| **Rovinj** | 0.548 | 0.749 | 84 | 0.597 | 0.728 | 58 |





**Figure 10: Box-plots of the SSIM values at each day of the testing period, in dependence on the assigned cluster. Plots are for each station for a choice: (left pane) two, and (right pane) three clusters. Black dots indicate dates of the testing period which contain episodes of extreme HF oscillations (both HF and Compound extremes). In box-plots, orange lines stand for median values, while the upper and lower edges of blue boxes denote the 75th and 25th percentile value, respectively. Whiskers denote the minimum and maximum values.**

## 3.4 Method selection

There are many methods available for clustering data, such as k-mean, self-organizing maps (SOMs), k-medoid, etc. We have chosen the k-medoids with the SSIM as the distance measure. In this section we clarify or reasoning on the choice made.



To find characteristic synoptics fields that are related to certain process (in this case intense HF sea level oscillations) both subjective and objective approaches can be implemented. The subjective approach requires an observer who manually classifies the synoptic situations based on previous experience and gained knowledge (which can be only manual as in Muller (1977) or manual after some programming filter as in Sheridan (2002)). This approach is useful for a

qualitative analysis and was already applied to problem at hand by Ramis and Jansà (1983), Šepić et al., (2015a), Bechle et al. (2016), Pellikka et al. (2022), etc. However, this approach becomes more difficult and time consuming as the data set gets larger. An objective approach, i.e., mathematical methods for classification can also be used for classification, thus diminishing the big data set problem. The most popular methods for classification are k-means, SOM, and herein presented k-medoid method, which have been used to classify various problems in atmosphere physics such as the North Atlantic

climate variability (Reusch et al., 2007), synoptic and local-scale wind patterns in Tyrrhenian coastal area (Di Bernardino et al., 2022), reconstruction of ocean surface temperature and salinity (Elken et al., 2019) and many other. Additionally, the empirical orthogonal function (EOF) analysis or principal component analysis (PCA) are used by various authors, but as a dimensionality reduction technique which precedes one of the clusters methods (Philippopoulos et al., 2014; Gao et al., 2019).

The k-means and SOM classification methods are known to generate good results but have some underlying problems which need to be kept in mind. Firstly, the centres of clusters are mean values of each cluster element, potentially clouding the physical processes that are driving the events in each cluster and resulting with a loss (due to averaging) of some valuable information about the synoptic situation. The commonly used distance metric in k-means and SOM is the Euclidian distance which takes point to point distances, occasionally leading to misleading results e.g., sometimes, a pressure

low separated by only a few points, in two events, will result with a large Euclidian distance, although physically the two situations are very similar (we are more interested in gradients associated to the low, than to the exact value of the low). On top of that, regarding this particular problem, a low number of events in the training set (from 30 to 56 (sum of HF and Compound), Table 1.) can represent a problem for k-means and SOM algorithms. These issues can be somewhat overcome by using different normalizations on the input data which help improve the results (Milligan and Cooper, 1988). An extra

problem arises when the input data is chosen to be a compound of different variables. This enlarges the dimensionality of input data and requires a lot more events for SOM and k-means to be trained effectively (Wang et al., 2019).

The issue of large dimensionality and small number of training events can be somewhat resolved by using the k-medoid algorithm that choses specific events (medoids) rather than average values, to represent clusters. K-medoid allows for the better representation of the physical process which are in the background of each even and which can be better

understood by examining the characteristic medoid (Winderlich et al., 2024). The problem of the distance metric is amended by using the structural similarity index measure (SSIM) which treats input fields as images (Hoffmann et al., 2021) ensuring that similar synoptic situations get grouped accordingly.

The test which of the three mentioned methods (k-means, SOM and k-medoid) is optimal for our problem, we used all three, separately, to find clusters of the training set, and then to label (associate to a cluster) all days in the testing period



(comparison of three methods is shown in Fig. 12 for Bakar). For k-means and SOM, Euclidian distance was used as a distance metric, and for k-medoid method SSIM was used. In Fig. 12, black dots represent days in the testing period in which an extreme event occurred (HF or Compound). The y-axis has the normalized distances, i.e., the smaller the value the closer (more similar) is the day to its cluster center (for SOM and k-means) or to its medoid (for k-medoid). We notice that for a choice of k-medoid method almost all events (black dots) happen when the normalized distance is small in comparison

with the other two methods, regardless of our choice of number of clusters. For k-means method, both for the choice of two and three clusters, the events happen with no clear connection to normalized distance. The same thing can be said for SOM which has, for one cluster, events that happened when the normalized distance was extremely large and events for which it was extremely low. Conclusively, we can say that out of the three tested methods, k-medoid works the best in our situation.



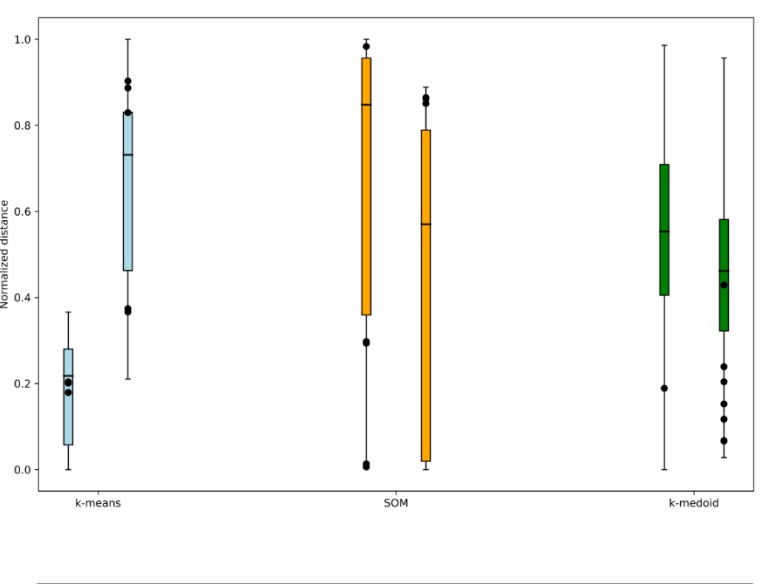

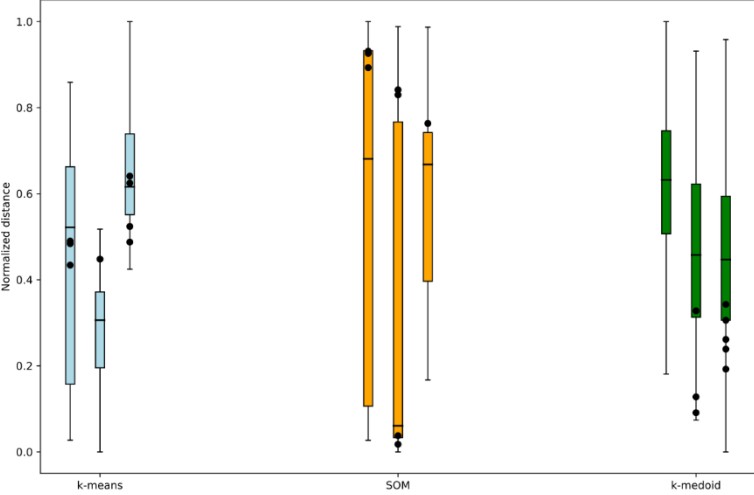


**Figure 11: Box-plots of normalized distances between each day of the testing period and the assigned cluster for a choice of (upper plot) two and (bottom plot) three clusters. Black dots represent the days in the testing period when HF or Compound extreme events occurred. The analysis was carried out for three methods, k-means, SOM and k-medoid for tide gauge Bakar.**

## 4 Discussion and conclusions

460        Traditionally, HF sea level extremes in the Adriatic, but also in the Mediterranean, have been associated with a single synoptic pattern characterized by a non-gradient (uniform) pressure field over the Adriatic, an inflow of warm air from the southwest in the lower troposphere, and a presence of a front side of a deep trough (with strong south-westerly





winds over the Adriatic) in the mid and high troposphere (Jansà et al., 2007; Vilibić and Šepić, 2009; Šepić et al., 2009; Šepić et al., 2015a; Šepić et al., 2015b). The work presented here led to the detection of a similar pattern (Cluster B1; Figs.

6-8 and S1-3) which was mostly associated to HF extremes which occurred from June to September (Fig. 9), and which we labelled as "summer-type" pattern. However, we detected at least one, and at some stations two additional synoptic patterns that were clearly different from the "summer-type" pattern, and which were present over the Adriatic during other HF and Compound extremes. The second distinct pattern (Cluster B3; Figs. 6-8 and 8 S1-3) was characterized by pronounced MSLP gradients over the Adriatic, leading to strong south-easterly winds at the surface, an inflow of warm air in the lower

troposphere (with the atmosphere generally cooler than during summer events) and, again, a presence of a front side of a deep trough in the mid and upper troposphere. This pattern was mostly assigned to HF and Compound extremes which occurred during colder half of the year (Fig. 9), and we have thus labelled it "winter-type" pattern. Summer- and winter-type patterns mostly differed at surface, while they were more similar at higher altitudes. The differences in surface weather were reflected in differences in the LF component of sea level. The summer-type pattern, where there were no strong MSLP

gradients and winds, was associated with average or slightly higher than average background (LF) sea level heights (Figs. 6-8, S1-3 fourth row). On the other hand, the winter-type pattern, where the MSLP was lower-than-average with pronounced gradients, and strong south-easterly winds were associated with above average background (LF) sea levels (Figs. 6-8, S1-3; fourth row). The distinction also resulted in the summer-type pattern being more typical of HF extremes, and the winter-type pattern being more typical of Compound extremes. The herein extracted winter-type pattern is clearly similar to storm surge

favourable pattern as recognized for the Adriatic Sea by e.g., Lionello et al. (2012). Finally, we have also extracted an additional synoptic pattern (Cluster B2 in Figs. 6-8, S1-3). At most of the stations this is a transitional-type pattern and presents a refinement of either the summer- or winter-type pattern. However, at Bakar and Rovinj station, the additional pattern (Cluster B2; Fig. 6, Fig. 1S) described a new situation with pronounced MSLP gradients, but such that moderate to strong north-easterly winds (i.e., Bora, Grisogono and Belušić, 2009) were blowing over the area, suggesting that additional,

previously unrecognized types of weather systems may lead to strong HF sea level oscillations along some coastal stretches.

The extraction of two additional synoptic patterns related to HF sea level oscillations represents a novelty of our work, as so far only the summer-type pattern has been recognized as characteristic for the occurrence of HF sea level oscillations in the Adriatic and Mediterranean Sea (aside for a few recent exceptions such as Pérez-Gómez et al, 2021; Ferrarin et al., 2021). We particularly stress the recognition of the winter-type pattern which is often found over the area

during Compound extremes, i.e., sea level extremes in which both the HF and LF components (the latter usually as a storm surge) contribute jointly to the overall height of the extreme. The importance of evaluating the HF sea level oscillations when assessing the total flooding heights during a storm surge has only recently been acknowledged (Pérez-Gómez et al., 2021; Heidarzadeh and Rabinovich, 2021; Medvedev et al., 2022; Rabinovich et al., 2023), and our work represents an important contribution to such an assessment by confirming that HF sea level extremes can occur during synoptic situations

which favour development of storm surges.



Although the primary aim of our work was to extract and classify synoptic patterns characteristic for HF and Compound extremes in the Adriatic Sea, our results raise an important question on applicability of the k-medoid algorithm to forecasting HF and Compound extremes. Evaluation of synoptic fields with an aim of forecasting "rissagas" (local name for meteotsunamis) has been done for the Balearic Islands since 1985 (Jansà and Ramis, 2021) and has been based on the

premise that the greater the similarity (visually assessed) of forecasted synoptic fields to one characteristic synoptic situation, the higher the probability of a strong meteotsunami. Šepić et al. (2016) suggested a method for quantifying the Balearic "rissaga" forecast by proposing a meteotsunami synoptic index that associates the average height of short-period sea level oscillations in Ciutadella (Balearic Islands) to a linear combination of selected synoptic variables (e.g. wind speed and direction in the low and mid troposphere, gradients of MSLP and temperature, etc.). Both the probabilistic forecast (Jansà

and Ramis, 2021) and a meteotsunami index (Šepić et al., 2016) have proven successful in predicting "no-rissaga/rissaga" conditions – if there is no favourable synoptic pattern, there will be no "rissaga", if there is a pattern, there could be a "rissaga" - but both have been less successful in predicting "rissaga" strength. Additionally, both are strictly related to one characteristic synoptic situation.

Unfortunately, when it comes to forecast, our classification k-medoid algorithm suffer from the same problems as

the two methods applied to the Balearic Islands. Ideally, we would say that HF sea level oscillations will be greatly amplified when the synoptic situation over an area is such that it strongly resembles one of the herein extracted synoptic patterns. However, as Figs. 10 and 11 reveal, we are still a long way from reaching this conclusion. At most stations, most of the extremes during the test periods, did occur at times when the synoptic conditions were similar to the extracted patterns (SSIM above the 75th percentile). However, there are obviously many other days when the synoptic conditions were as

similar (and even more similar) to the extracted patterns, but on which HF or Compound extremes were not registered. It seems that our analysis once again confirms the conclusions of Šepić et al. (2016) that the presence of favourable synoptic conditions is a necessary but not sufficient condition for the strong amplification of high-frequency sea level oscillations, as synoptic maps "miss" the mesoscale atmospheric disturbances that actually generate intense high-frequency sea level oscillations.

Nevertheless, the k-mediod classification method could be used to complement other methods for forecasting extreme HF sea level oscillations. For example, in situations where one of the specific patterns is observed/predicted (i.e., in situations where there is a high SSIM score between the predicted synoptic field and one of our clusters), additional attention could be paid to higher-resolution atmospheric and ocean models (as described, for example, in Denamiel et al., 2019; Mourre et al., 2021), neural network methods (e.g., Vich and Romero, 2021), and real-time air pressure and sea level

measurements (e.g., Marcos et al., 2009). Here we stress that it is precisely the combination of different elements (synoptic assessment, high-resolution modelling and real-time measurements) that has been recognised as a necessity for the development of an efficient system for monitoring and forecasting meteorological tsunamis (as the most destructive type of atmospherically triggered HF sea level oscillations) (Vilibić et al., 2016). We conclude by saying that our work contributes



to this worthy goal by exploring a novel method for classifying characteristic synoptic fields and by demonstrating that
extreme HF sea level oscillations occur under more than one typical weather.

**Author contribution**

All of the authors helped with the conceptualization of the paper. MV preformed the initial manual extraction of characteristic synoptic situations, KR did the latter data analysis, JS and KR wrote the manuscript while KR, JS and MV edited and corrected the text. JS supervised the analysis and the writing of the manuscript.

**Competing interests**

The authors declare that they have no conflict of interest.

**Acknowledgements**

We thank the Department of Geophysics, Faculty of Science, University of Zagreb and Hydrographic Institute of Republic of Croatia, for diligently taking care of tide gauge stations Rovinj, Bakar, Zadar, Split, Ploče and Dubrovnik, and
providing high quality sea level data. Furthermore, we thank Hrvoje Kalinić, Leon Ćatipović (Faculty of Science, University of Split) and Frano Matić (University of Split) for fruitful discussions related to selection of classification algorithm. Lastly, we acknowledge the ERC StG 853045 SHExtreme and HRZZ IP-2019-04-5875 StVar-Adri projects for supporting the research.



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
