# Peer review of "Synoptic patterns associated with high-frequency sea level extremes in the Adriatic Sea"

_EGUsphere, 2024_

## Author Response (AR1)

**Reviewer #1**

We thank the reviewer for a thorough review of our manuscript and for positive and constructive comments. We will address all comments, make the suggested corrections and add the necessary additional explanations to the manuscript.

**Specific comments:**

**Comment:** Line 207: Don't need to include, just curiosity on my part but any thoughts on why the events are more abundant between 2016 and 2021 at Rovinj?

**Answer:** Thank you for raising this important question and drawing our attention to this issue. Following your comment, we have contacted the provider of the sea level data, the Hydrographic Institute of the Republic of Croatia. Apparently, there was a change in the characteristics of the stilling well in 2016 and 2017. The connecting pipe first broke during a storm in 2016 and was then replaced in 2017 with a new connecting pipe with a different shape and different attenuation properties (compared to the original pipe). Both changes probably resulted in weaker damping of the high-frequency oscillations in the stilling well. This is now noted in the manuscript. Lines 228-234.

Despite the changed properties, we decided to keep the Rovinj station in our manuscript (and in the supplementary figures) because: (1) more than half of the extreme events detected at Rovinj in the period 2016-2021 were also detected at other tide gauge stations; (2) the k-medoid analysis provided results that are consistent with the results for other stations, especially with the results for the closest station in Bakar Bay.

**Comment:** The conclusion may benefit from bullet points summarising the key findings of the work.

**Answer**: We agree, we added bullet points. Lines 566-591.

**Comment:** Again, just curiosity on my part, but how did the testing period fair against the training period? Did it meet your expectations? I know it is difficult to say given the small number of events, maybe a better question should be do you expect it to perform to trend in the future with a longer-term dataset?

**Answer**: We are partially satisfied. Almost all "extreme days" in the test period had high similarity scores - i.e. the synoptics resembled those of the characteristic medoids. Nevertheless, the predictive potential appears to be relatively low. Following your comments, we will list points that could help to improve the methodology, including the suggested extension of the training set.

**Technical corrections:**

**Comment:** Line 32: Spell check 'centimeter'

**Answer:** Corrected.

**Comment:** Line 200: Fig 3, move a) and b) on graphs to be more easily seen and maybe reword the latter part of the description

**Answer:** We moved  a) and b) on the graphs and rewrite the second part of the caption.

**Comment:** Line 273-4: Punctuation needed "Bora; Grisogono"

**Answer:** Corrected.

**Comment:** Line 298-99: Spell check "isobars"

**Answer:** Corrected.

**Comment:** Line 338: Should A2 be both a transitional and a winter type event?

**Answer**: Indeed not. Thank you for noticing this. A2 should just be "winter-type". We will correct it.

**Comment:** Line 409: "or" should be "our"

**Answer:** Corrected.

**Comment:** Line 415 and 417: Reword as a little confusing

**Answer:** Corrected.

**Comment:** Line 421: "other" should be "others"

**Answer:** Corrected.

**Comment:** Line 423: "clusters" should be "cluster"

**Answer:** Corrected.

**Comment:** Line 426: Reword as a little confusing

**Answer:** We rewrote the sentence.

**Comment:** Line 439: "even" should be "event"

**Answer:** Corrected.

**Reviewer #2**

We thank the reviewer for a thorough review of our manuscript and for positive and constructive comments. We will address all comments, make the suggested corrections and add the necessary additional explanations to the manuscript.

**Comment:** I find particularly important in this study the emphasis on the compound events. Despite it is well justified by the authors that the focus of this work is on the combination of meteotsunamis and storm surges

**Answer:** Thank you!

**Comment:** I would find interesting a brief mention to other potential contributors to total sea level such as the tide or the seasonal cycle, either in the introduction or in the discussion. Perhaps a brief description of the tide characteristics and amplitude in this area could be added.

**Answer:** We added (in the introduction, lines 71-82) additional information on the processes leading to the extreme sea levels of the Adriatic, especially tides, but also other longer period changes/oscillations, including seasonal signal, annual, decadal and climate scale changes.

**Comment:** Information about which of the stations used could be presenting amplification of sea level oscillations due to more local effects such as harbour resonance would also be interesting.

**Answer:** We added this information in our Discussion section. Lines 526-536.

**Comment:** Finally, I recommend to include in the discussion the convenience of future studies which could better identify the contribution of waves (infragravity waves), the limitations of 1-min sampling for this objective, and recommendations to solve this in the implementation of sea level networks.

**Answer:** We agree. We included a few lines about the contribution of wind waves, swell and infragravity waves to the Adriatic sea level extremes into Introduction (lines 61-65). In the discussion we addressed the importance of including higher sampling rate measurements in sea level networks. This is shown in the last bullet point of the conclusion. Lines 589-591.

**Comment:** Do the authors have an explanation for the increased number of events in Rovinj between 2016-2021? This kind of analysis will become particularly important, as the time series grow, to analyse potential changes driven by climate change.

**Answer:** Thank you for raising this important question and drawing our attention to this issue. Following your comment, we have contacted the provider of the sea level data, the Hydrographic Institute of the Republic of Croatia. Apparently, there was a change in the characteristics of the stilling well in 2016 and 2017. The connecting pipe first broke during a storm in 2016 and was then replaced in 2017 with a new connecting pipe with a different shape and different attenuation properties (compared to the original pipe). Both changes probably resulted in weaker damping of the high-frequency oscillations in the stilling well. This is now noted in the manuscript. Lines 228-234.

Despite the changed properties, we decided to keep the Rovinj station in our manuscript (and in the supplementary figures) because: (1) more than half of the extreme events detected at Rovinj in the period 2016-2021 were also detected at other tide gauge stations; (2) the k-medoid analysis provided results that are consistent with the results for other stations, especially with the results for the closest station in Bakar Bay.

In view of this, we consider the series too short to discuss the effects of climate change. Nevertheless, we agree that it is important to assess the change in high-frequency extremes in a changing climate – and for this reason to ensure the continuation of high-resolution sea level measurements. We emphasise this in the discussion.

**Technical corrections**

**Comment**: Line 165: typo: "To determine medoids"

**Answer**: Thank you, corrected.

**Comment:** Line 439: typo: " in the background of each event"

**Answer:** Corrected